ⓐ | **Open Peer Review** | Clinical Microbiology | Methods and Protocols

# Enhancing *Acanthamoeba* diagnostics: rapid detection of viable *Acanthamoeba* trophozoites and cysts using viability PCR assay

J. M. J. Veugen,[1,2,3,4] P. H. M. Savelkoul,[3,4] R. M. M. A. Nuijts,[1,2,5] M. M. Dickman,[1,2] P. F. G. Wolffs[3,4]

**ABSTRACT** *Acanthamoeba* keratitis (AK) is a sight-threatening corneal infection that is challenging to diagnose and treat due to the resistance of *Acanthamoeba* to standard antimicrobial agents. Current detection methods have limitations. This study aimed to develop and validate a sensitive viability PCR (v-PCR) assay using a photoreactive dye to distinguish viable from non-viable *Acanthamoeba* for rapid identification of viable *Acanthamoeba* trophozoites and cysts. Propidium monoazide (PMAxx) was used as a photoreactive dye. Mixtures containing decreasing percentages of viable *Acanthamoeba*, including reference strains *Acanthamoeba polyphaga* trophozoites and cysts, *Acanthamoeba castellanii* trophozoites, and *Acanthamoeba castellanii* trophozoites from a clinical sample, were prepared. Disinfectant efficacy against *Acanthamoeba* was also assessed. Samples were divided into PMAxx-treated and non-PMAxx-treated parts, and v-PCR assay was applied to both. The difference in viable *Acanthamoeba* was determined by subtracting the cycle threshold (Ct) value of the PMAxx-treated sample from the non-PMAxx-treated sample. Mixtures with decreasing concentrations of viable *Acanthamoeba* trophozoites and cysts showed increasingly lower delta Ct values as the percentage of viable *Acanthamoeba* decreased, as expected. This relationship was observed across all tested samples. Menicon Progent effectively eliminated *A. polyphaga* trophozoites and cysts, while propamidine, chlorhexidine, or their combination resulted in approximately 2-log reductions in *A. polyphaga* trophozoites and cysts. In the current study, a rapid v-PCR assay was developed that can distinguish between viable and non-viable *Acanthamoeba*, for both trophozoites and cysts, across multiple species. The presence of viable *Acanthamoeba*, as determined by v-PCR, allows monitoring of treatment response and efficacy in AK.

**IMPORTANCE** The development of a sensitive viability PCR (v-PCR) assay using propidium monoazide (PMAxx) as a photoreactive dye marks a significant advancement in the diagnosis and treatment of *Acanthamoeba* keratitis (AK), a severe corneal infection notorious for its resistance to conventional antimicrobials. This innovative assay offers a rapid and accurate method to distinguish viable from non-viable *Acanthamoeba* trophozoites and cysts, addressing a critical need in the field. By effectively distinguishing between viable and non-viable *Acanthamoeba*, this test enables monitoring of treatment response and efficacy, essential for guiding clinical interventions in AK cases. The successful validation of this v-PCR assay across various *Acanthamoeba* species and its ability to assess disinfectant efficacy further underline its potential as a valuable tool for improving diagnostic precision and therapeutic outcomes in the treatment of AK.

**KEYWORDS** *Acanthamoeba*, *Acanthamoeba keratitis*, viability PCR, PMAxx, disinfectant efficacy

**Peer Reviewers** Pieter W. Smit, Maasstad Hospital, Rotterdam, the Netherlands; Mehmet Aykur, Ege University Parasitology, Bornova, Izmir, Turkey

Address correspondence to P. F. G. Wolffs, p.wolffs@mumc.nl.

The authors declare no conflict of interest.

A*canthamoeba* includes a diverse group of microorganisms found worldwide in various environments, such as soil, freshwater, and marine habitats, as well as in in

dust and air (1, 2). *Acanthamoeba* are single-celled, free-living amoebae that exist in two forms: the active trophozoites that move through contractions of finger-like projections called acanthopodia, and the dormant cysts, encapsulated in double-layered cellulose walls. The cyst form is highly resilient and can withstand extreme environmental conditions, allowing transmission to occur under adverse circumstances (1, 2). While the majority of *Acanthamoeba* species are harmless and fulfill ecological roles (3, 4), certain species within the genus have gained attention for their ability to cause infections in humans and animals (5). These opportunistic parasites can lead to serious conditions, including *Acanthamoeba* keratitis (AK) and granulomatous amoebic encephalitis, which present significant challenges to human health (1, 3, 6–8). The most common *Acanthamoeba* species associated with keratitis are *Acanthamoeba castellanii* and *Acanthamoeba polyphaga*. These two species are responsible for the majority of reported cases of AK worldwide (2, 9).

AK is a severe corneal infection that is often extremely painful and poses a significant threat to vision. Most AK cases are associated with the use of soft contact lenses and poor contact lens hygiene (2, 9). However, ocular trauma involving contaminated water or soil also poses a risk factor (9). AK causes corneal damage and can potentially lead to blindness. Diagnosing and treating this infection can be challenging due to the resistance of *Acanthamoeba* cysts to many standard antimicrobial agents (10–12).

AK requires a prolonged treatment duration and involves the intensive application of topical disinfectants. The primary treatment options for AK typically include polyhexamethylene biguanide (PHMB), propamidine (Brolene), chlorhexidine, or a combination of these agents (1, 11–14). Systemic medications may be used as adjunctive therapy (1, 12). In non-responsive cases, therapeutic penetrating keratoplasty may be necessary (1, 3, 10). The prognosis of AK depends on the severity at presentation. Delayed diagnosis and treatment can lead to poor visual outcomes, including corneal scarring and permanent opacification (9, 15). Long-term follow-up is necessary due to the potential reactivation of dormant cysts. Early diagnosis, coupled with aggressive treatment, offers the best chance of cure (3).

Currently, AK is diagnosed using culture or PCR. Culture-based methods are time-consuming, complex, and have impaired sensitivity compared with molecular techniques (15). Moreover, cyst forms may not excyst, or trophozoites may become damaged during sample preparation, leading to undetected infections (16). Real-time polymerase chain reaction (PCR) is an attractive alternative to culture-based methods due to its increased sensitivity and specificity (16–19), but this technique cannot distinguish between viable and non-viable *Acanthamoeba*.

The timely and accurate diagnosis of AK is of vital importance for initiating appropriate treatment and infection control guidance. A positive PCR result indicating the presence of *Acanthamoeba* can have significant implications for treatment decisions. However, without the ability to distinguish between viable and non-viable *Acanthamoeba*, a positive PCR result alone may not provide sufficient information to effectively guide treatment. Ophthalmologists play a crucial role in evaluating the effectiveness of the treatment regimen to ensure optimal management of AK.

Viability PCR (v-PCR) offers a potential solution, wherein samples are pre-treated with a photoreactive dye, like propidium monoazide (PMA). PMA selectively binds to free nucleic acid (NA) or NA of non-viable *Acanthamoeba*, preventing subsequent NA amplification and detection via PCR (20–26). This approach enables only the specific amplification and identification of DNA from viable *Acanthamoeba*, which is particularly valuable in cases where initial therapy does not yield satisfactory results. This underlines the importance of obtaining accurate information to guide subsequent treatment decisions and improve patient outcomes. The aim of this study was to develop and validate a sensitive v-PCR assay, which is rapid and easy to perform, that can distinguish between viable and non-viable *Acanthamoeba* trophozoites and cysts across multiple species.

## MATERIALS AND METHODS

### *Acanthamoeba* trophozoites

The *Acanthamoeba* reference strains were obtained from the American Type Culture Collection (ATCC).

*A. polyphaga* ATCC strain 30461 (LGC standards GmbH, Wesel, Germany) trophozoites were cultured in axenically conditions using ATCC medium 712 in 75 cm$^2$ tissue culture flasks (Greiner Bio-one B.V., Kremsmünster, Austria) at 25°C. Subculturing of amoebae was performed when the amoebae formed a nearly continuous monolayer of cells on the bottom surface of the tissue culture flask.

*A. castellanii* ATCC strain 30868 (LGC standards GmbH, Wesel, Germany) trophozoites were cultured in xenic conditions using ATCC Page's Amoeba Saline (PAS) medium 1323 in 75 cm$^2$ tissue culture flasks at 25°C. Additionally, trophozoites from a clinical sample identified as *A. castellanii* were cultured in the same xenic conditions at a temperature of 30°C.

For both cultures, *Escherichia coli* ATCC strain 8739 was used as a food source. Therefore, *E. coli* was cultured on a blood agar plate (Becton Dickinson GmbH, Heidelberg, Germany) and then resuspended in PAS medium to achieve a final concentration of $1.5 \times 10^7$ colony-forming units per milliliter (CFU/mL).

To obtain an enriched trophozoite culture, subculturing of *Acanthamoeba* was performed every 2–3 days. Prior to subculturing, trophozoites were washed three times with phosphate-buffered saline (PBS) at pH 7.2 (Gibco, Thermo Fisher Scientific, Waltham, MA) and recovered by centrifugation at 300×*g* for 5 min. The washed cells were resuspended in 18 mL of PAS medium containing fresh *E. coli*.

### *Acanthamoeba* cysts

*A. polyphaga* ATCC strain 30461 cysts were obtained by transferring the trophozoites to PAS medium in 75 cm$^2$ tissue culture flasks at 25°C. The *Acanthamoeba* trophozoites were washed three times with PBS and centrifuged at 300×*g* for 5 min, the washed cells were resuspended in 18 mL of PAS medium. The encystation process of the *Acanthamoeba* was monitored by microscopy.

### Harvesting *Acanthamoeba* trophozoites and cysts

*Acanthamoeba* trophozoites and cysts were harvested by centrifugation at 300×*g* for 5 min. After centrifugation, *Acanthamoeba* was resuspended in PAS medium and quantified using a Neubauer hemocytometer chamber. Subsequently, *Acanthamoeba* was diluted in PAS medium to achieve a final concentration of $10^5$ cells/mL for the experiments.

### *Acanthamoeba* inactivation and sample preparation

#### *Autoclave inactivation treatment*

To create mixtures with varying concentrations of viable *Acanthamoeba*, *Acanthamoeba* at a concentration of $10^5$ cells/mL solutions were divided into two parts, a viable part and a non-viable part. The non-viable parts were inactivated by subjecting trophozoite and cyst cultures from *A. polyphaga*, as well as trophozoite cultures from *A. castellanii* and a strain identified as *A. castellanii* from a clinical corneal tissue sample, to autoclaving at 121°C for 15 min. Subsequently, the non-viable parts were mixed with the viable parts to generate mixtures with concentrations of 100%, 50%, 10%, 1%, 0.1%, and 0% viable *Acanthamoeba*.

In addition, 500 µL of the inactivated trophozoites and cysts was cultured on a non-nutrient agar plate supplemented with *E. coli* ATCC 8739 for a duration of 1 week. The growth of *Acanthamoeba* was monitored daily by microscopy throughout the incubation period to assess the presence of both trophozoites and cysts. This served as a control for the effectiveness of the autoclave inactivation treatment.

The autoclave inactivation treatment proved to be ineffective in fully eliminating viable *Acanthamoeba* cysts. In all experimental samples, both viable *Acanthamoeba* cysts and viable *Acanthamoeba* encysted trophozoites were detected using PMA-PCR. These results were confirmed when the autoclaved trophozoites and cysts samples of the different *Acanthamoeba* strains were cultured. Sporadically, visible *Acanthamoeba* cysts were observed under microscopy immediately after culturing, followed by sporadic identification of *Acanthamoeba* trophozoites after 3 days of incubation. These results indicate that the autoclave treatment used for inactivating *Acanthamoeba* trophozoites and cysts eradicated the majority of *Acanthamoeba* but could not completely eliminate all viable organisms.

To assess the impact of the ineffective autoclave inactivation treatment, the experiments were repeated using naked *Acanthamoeba* DNA. The results obtained with naked *Acanthamoeba* DNA were comparable to the results obtained with the autoclave inactivation treatment.

### Efficacy of disinfectants

The efficacy of Menicon Progent intensive cleaner, which is a protein-removing and disinfectant solution for hard lenses, was tested. Additionally, the efficacy of propamidine isetionate 0.1%, chlorhexidine digluconate 0.02%, and a combination of both propamidine isetionate 0.05% and chlorhexidine digluconate 0.01% were tested, as these topical medications are commonly used as the firstline treatment options. Both trophozoite cultures and cyst cultures from *A. polyphaga* were resuspended in the appropriate disinfectant at a concentration of $10^5$ cells/mL and incubated for 30 min, following Menicon Progent's manufacturer's instructions.

### PMA treatment

Samples were pre-treated with the photoreactive dye PMAxx (Biotium, Inc., Hayward, CA) to prevent amplification from free accessible DNA or DNA from non-viable *Acanthamoeba*. Each mixture containing a different concentration of viable *Acanthamoeba* was divided into two tubes: one with PMAxx added (+PMA) and one without PMAxx (−PMA). PMAxx was added to the +PMA tubes at a final concentration of 100 μM for both trophozoites and cysts. The +PMA tubes were vortexed and incubated in the dark at room temperature (RT) for 30 min. Following incubation, the +PMA tubes were subjected to photolysis for 15 min using the PMA-Lite system (Biotium, Inc., Hayward, CA) (Fig. 1).

### Nucleid acid isolation

All samples were analyzed at the department of Medical Microbiology, Infectious Diseases & Infection Prevention, Maastricht University Medical Center+, Maastricht, the Netherlands. DNA was extracted using the MagNA Pure 96 system (Roche Diagnostics GmbH, Mannheim, Germany). Extraction was performed using the MagNA Pure 96 DNA and Viral NA Small Volume Kit (Roche Diagnostics GmbH, Mannheim, Germany) and the Pathogen Universal 200 Protocol (MagNA Pure 96 system, Roche Diagnostics). Then, 100 μL sample was extracted and eluted in 50 μL elution buffer and diluted with 50 μL water for molecular biology (VWR International, Radnor, PA).

### PCR analysis

PCR assays were carried out on a Quantstudio five system (Applied Biosystems, Thermo Fisher Scientific, Waltham, MA) using a viability assay targeting the 18S ribosomal RNA (rRNA) gene. The forward and reverse primer sequences for the v-PCR were F400 5′-CCCAGATCGTTTACCGTGAA-3′ and R400 5′-AATATTAATGCCCCCAACTATCC-3′, respectively. The probe sequence was P200 5′-6-FAM- CTGCCACCGAATACATTAGCATGG-BHQ-1-3′ (18). Compared with the referenced publication, different concentrations were used, and the reverse primer was slightly modified. The final reaction volume was 20 μL and contained 10 μL 2× Taqpath BactoPure Microbial Detection Master Mix (Applied Biosystems, Thermo Fisher Scientific, Waltham, MA), 2 μL primer/probe mix, and 8 μL sample. Cycling

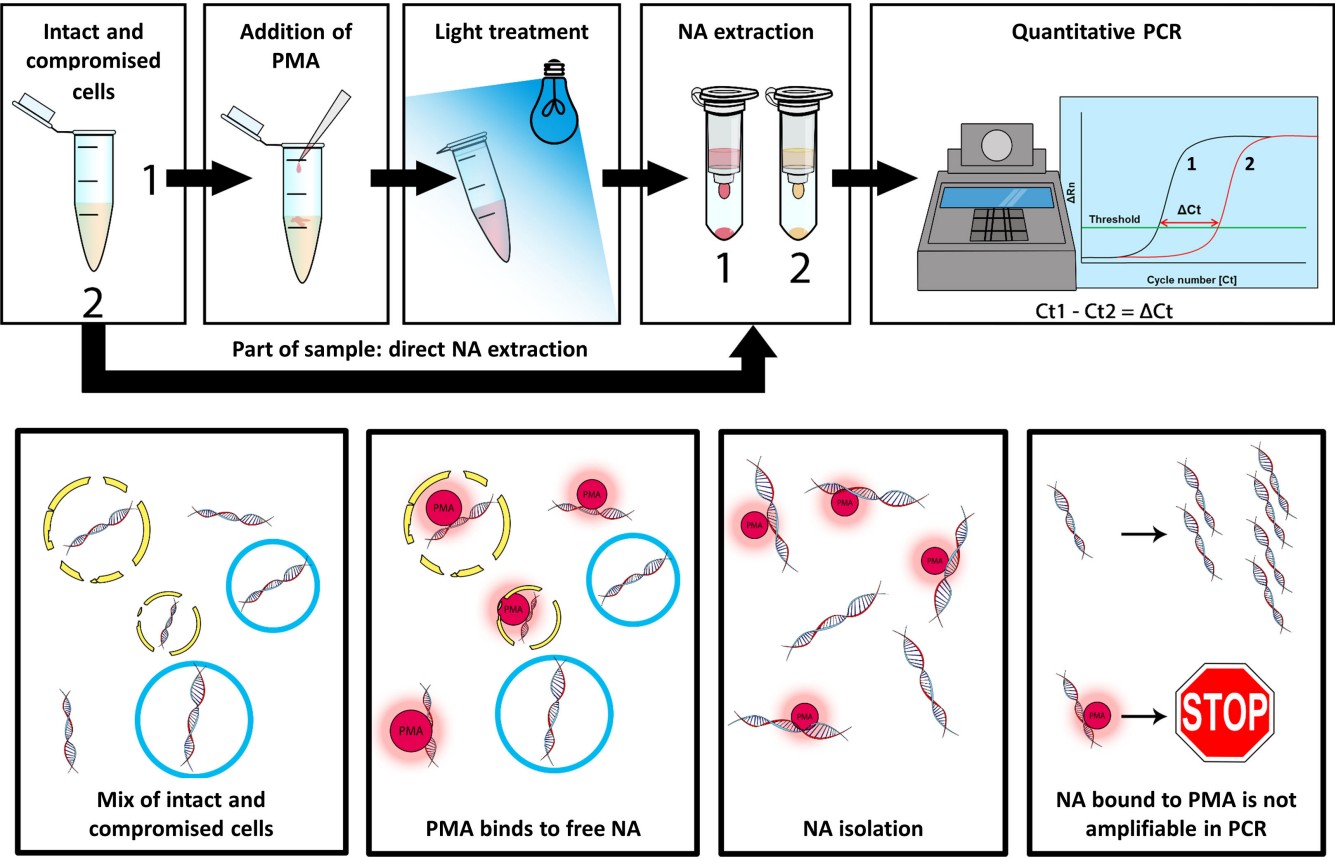

**FIG 1** Schematic representation of the viability PCR (v-PCR). Propidium monoazide (PMA) binds to free DNA or DNA of non-viable *Acanthamoeba*. DNA bound to PMA is not amplifiable in the PCR, while DNA of viable *Acanthamoeba* is unaffected by PMA and can be amplified with PCR. The delta cycle threshold (ΔCt) value is calculated by subtracting the Ct value of the PMAxx-treated sample from the Ct value of the non-PMAxx-treated sample.

conditions consisted of a pre-read step at 60°C for 30 s, initial denaturation/enzyme activation at 95°C for 2 min, and 42 cycles of denaturation at 95°C for 10 s and annealing/extension at 60°C for 30 s. The Ct value for each sample was determined using Quantstudio Design and Analysis Software v1.4.3.

## RESULTS

The amplification efficiency of the PCR reaction was 100% based on the standard curve, with a corresponding slope of −3.32 and $R^2$ of 0.98. Both cultured *Acanthamoeba* trophozoites and cysts of *A. polyphaga*, along with trophozoites of *A. castellanii* and a clinical sample identified as *A. castellanii*, were used to validate the ability of the v-PCR in distinguishing viable from non-viable *Acanthamoeba*. The validation involved analysis of mixtures with decreasing concentrations of viable *Acanthamoeba*, ranging from 100% to 0%. These serial dilutions, representing different ratios of viable *Acanthamoeba*, showed that the delta cycle threshold (Ct) values gradually increased as the ratio of viable *Acanthamoeba* decreased.

### *A. polyphaga* strain ATCC 30461 trophozoites and cysts

In cultured *A. polyphaga* trophozoites at a concentration of $10^5$ cells/mL, mixtures showed mean delta Ct values of −1.09 (100%), −0.77 (50%), −2.75 (10%), −5.69 (1%), −8.81 (0.1%), and −12.75 (0%). These values correspond to mean log reductions of 0.33, 0.23, 0.83, 1.71, 2.64, and 3.82, respectively. The serial dilution showed a linear correlation between the percentage of viable *Acanthamoeba* trophozoites (decreasing from 100% to

0% viable *Acanthamoeba* trophozoites) and the delta Ct value, $R^2 = 0.981$, as determined by linear regression on log-transformed values (Fig. 2A; Table S1).

In cultured *A. polyphaga* cysts at a concentration of $10^5$ cells/mL, mixtures showed mean delta Ct values of −1.72 (100%), −1.86 (50%), −4.41 (10%), −8.05 (1%), −9.80 (0,1%), and −10,82 (0%). These values correspond to mean log reductions of 0.51, 0.56, 1.32, 2.42, 2.94, and 3.25, respectively. The serial dilution showed a linear correlation between the percentage of viable *Acanthamoeba* cysts (decreasing from 100% to 0% viable *Acanthamoeba* cysts) and the delta Ct value, $R^2 = 0.981$, as determined by linear regression on log-transformed values (Fig. 2B; Table S1).

### *A. castellanii* strain ATCC 30868 trophozoites

In cultured *A. castellanii* trophozoites at a concentration of $10^5$ cells/mL, mixtures showed mean delta Ct values of −0.15 (100%), −0.48 (50%), −2.24 (10%), −5.61 (1%), −8.28 (0.1%), and −12.28 (0%). These values correspond to mean log reductions of 0.04, 0.14, 0.67, 1.68, 2.49, and 3.68, respectively. The serial dilution showed a linear correlation between the percentage of viable *Acanthamoeba* trophozoites (decreasing from 100% to 0% viable *Acanthamoeba* trophozoites) and the delta Ct value, $R^2 = 0.993$, as determined by linear regression on log-transformed values (Fig. 2C; Table S1).

### *A. castellanii* cultured from clinical sample

In cultured *A. castellanii* trophozoites at a concentration of $10^5$ cells/mL from a clinical sample, mixtures showed mean delta Ct values of −0.42 (100%), −0.22 (50%), −1.40

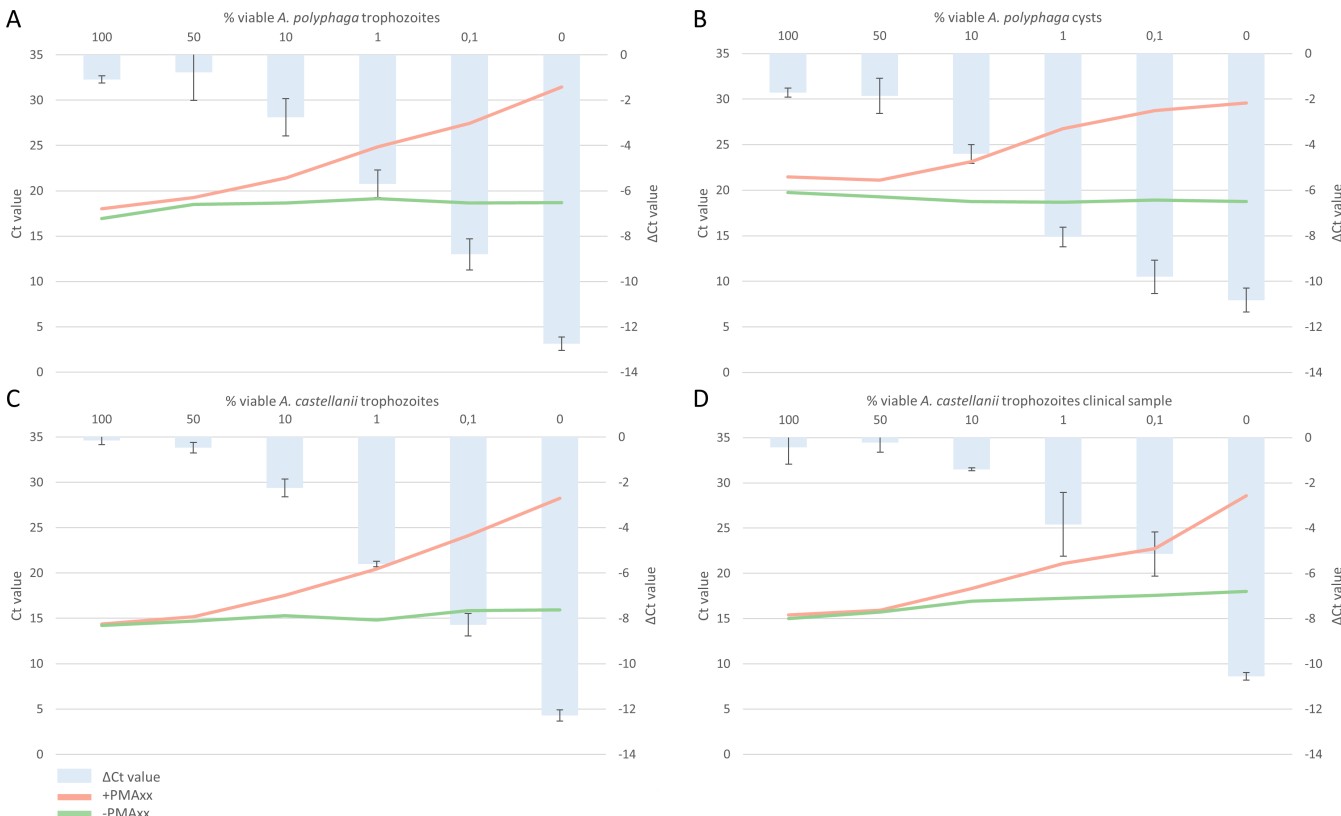

**FIG 2** Validation results of the v-PCR on (A) *A. polyphaga* trophozoites; (B) *A. polyphaga* cysts; (C) *A. castellanii* trophozoites; (D) *A. castellanii* trophozoites from a clinical sample. Mixtures with varying viability from 100% to 0% were divided into a PMAxx-treated vs non-PMAxx-treated samples. ΔCt values are calculated by subtracting the Ct value of the PMAxx-treated sample from the non-PMAxx-treated sample. Ct values are shown as means; ΔCt values are shown as means ± standard deviations; assays for *A. polyphaga* trophozoites and cysts, and *A. castellanii* trophozoites performed in triplicate, assays for *A. castellanii* trophozoites performed in duplicate. v-PCR, viability PCR; *A. polyphaga*, *Acanthamoeba polyphaga*; *A. castellanii*, *Acanthamoeba castellanii*; Ct, cycle threshold; ΔCt, delta cycle threshold.

(10%), −3.83 (1%), −5.14 (0.1%), and −10.56 (0%), respectively, corresponding to mean log reductions of 0.13, 0.07, 0.42, 1.15, 1.54, and 3.17, respectively. The serial dilution showed a linear correlation between the percentage of viable *Acanthamoeba* trophozoites (decreasing from 100% to 0% viable *Acanthamoeba* trophozoites) and the delta Ct value, $R^2 = 0.973$, as determined by linear regression on log-transformed values (Fig. 2D; Table S1).

## Efficacy of disinfectants

The assays conducted on both *A. polyphaga* trophozoites and cysts $10^5$ cells/mL in Menicon Progent for 30 min showed that Menicon Progent effectively eliminated the *A. polyphaga* trophozoites and cysts. Both the PMAxx-treated samples and the non-PMAxx-treated samples were unable to detect *Acanthamoeba* DNA using the PCR assay after a 30 min incubation period.

When *A. polyphaga* trophozoites at a concentration of $10^5$ cells/mL were treated with propamidine isetionate 0.1%, chlorhexidine digluconate 0.02%, and a combination of both propamidine isetionate 0.05% and chlorhexidine digluconate 0.01%, the mean delta Ct values were -8.23,–8.36, and −7.25, respectively (Fig. 3A). These values correspond to mean log reductions of 2.47, 2.51, and 2.18, respectively, after a 30 min incubation period.

Similarly, when *A. polyphaga* cysts at a concentration of $10^5$ cells/mL were treated with the same disinfectants, the mean delta Ct values were -6.36,–6.09, and −5.98, respectively (Fig. 3B). This resulted in mean log reductions of 1.91, 1.83, and 1.80, respectively, after 30 min of incubation.

## DISCUSSION

The purpose of this study was to validate a v-PCR assay that is both sensitive and rapid and is capable of distinguishing between viable and non-viable *Acanthamoeba* trophozoites and cysts across multiple species. The assay effectively discriminates between viable and non-viable *Acanthamoeba* by comparing the Ct values of PMAxx-treated samples and non-PMAxx-treated samples. In the present study, a sensitive, rapid, and easily implementable method for differentiating between viable and non-viable *Acanthamoeba* trophozoites and cysts has been established. To the best of our knowledge, this is the second study to utilize PMA in combination with PCR for differentiation of viable and non-viable *Acanthamoeba* species.

The findings from the current study show that the application of PMAxx treatment on cultured inactivated *Acanthamoeba* trophozoites and cysts resulted in a maximum mean

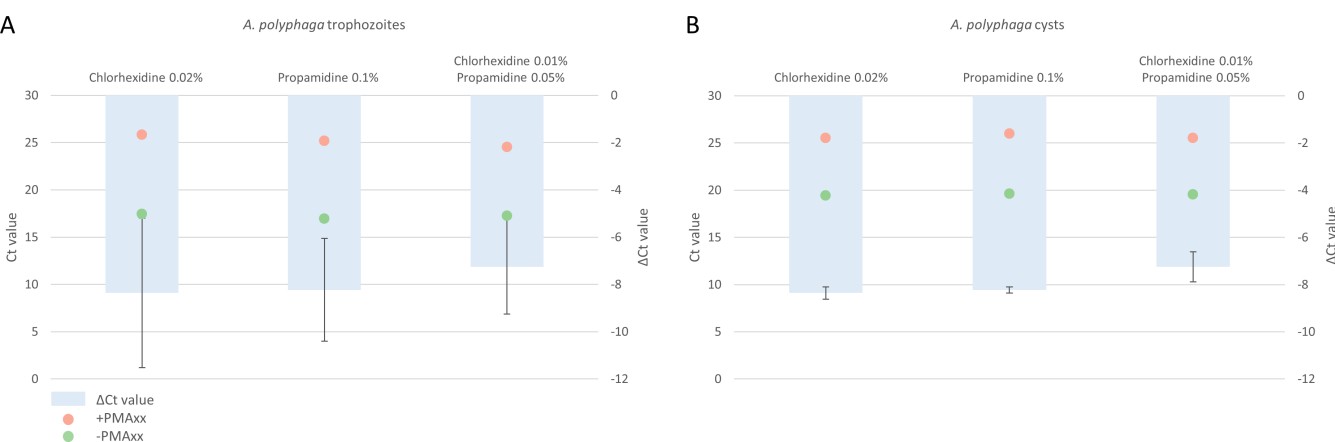

**FIG 3** Efficacy of propamidine isetionate 0.1%, chlorhexidine digluconate 0.02%, and a combination of propamidine isetionate 0.05% and chlorhexidine digluconate 0.01% against viable (A) *A. polyphaga* trophozoites and (B) *A. polyphaga* cysts. ΔCt values are calculated by subtracting the Ct value of the PMAxx-treated sample from the non-PMAxx-treated sample. Ct values are shown as means; ΔCt values are shown as means ± standard deviations; assays for *A. polyphaga* trophozoites and cysts performed in duplicate. *A. polyphaga, Acanthamoeba polyphaga*; Ct, cycle threshold; ΔCt, delta cycle threshold.

delta Ct value of −12.75 for trophozoites and −10.82 for cysts. These results indicate that PMAxx treatment effectively blocks nearly 99.99% of free DNA and DNA from non-viable *Acanthamoeba*. However, it is important to consider that the assay's sensitivity to detect small differences in viability, particularly at lower percentages of viable cells, may have implications for its clinical utility.

Conducting multiple studies to validate findings and assess reproducibility under different conditions is crucial. This approach ensures the reliability and generalizability of the results. The log reductions obtained in this study confirm comparable findings reported by Fittipaldi et al. for *Acanthamoeba* trophozoites and slightly lower for *Acanthamoeba* cysts. In their study, a 4-log reduction was achieved for *A. castellanii* trophozoites and cysts using PMA treatment with a concentration of 200 µM (25). However, we used PMAxx in our study, which is designed to more effectively inhibit PCR amplification of free DNA and non-viable DNA compared with PMA. A PMAxx concentration of 200 µM showed no increased effectiveness in blocking non-viable *Acanthamoeba* compared with 100 µM (data not shown). Interestingly, Fittipaldi et al. successfully inactivated *Acanthamoeba* by subjecting them to autoclaving at 121°C for 15 min, whereas autoclave inactivation at the same conditions in our study was insufficient. An advantage of our study is that we tested the v-PCR on two different *Acanthamoeba* ATCC strains, as well as on a clinically isolated *A. castellanii* strain.

As part of a clinical application, our results demonstrate the effectiveness of Menicon Progent intensive cleaner, a protein remover and disinfectant, against *Acanthamoeba* trophozoites and cysts. None of the experiments detected *Acanthamoeba* DNA after a 30 min incubation period. However, it is important to note that the use of Menicon Progent is limited to hard lenses. These findings align with previous research conducted to evaluate the efficacy of Progent against SARS-CoV-2 (27).

The topical disinfectants propamidine, chlorhexidine, and their combination, widely recognized as one of the primary treatment options for AK, demonstrated over 2-log reduction of viable *Acanthamoeba* trophozoites and nearly 2-log reduction of viable *Acanthamoeba* cysts. However, the combination was only tested at half the concentration used clinically. This may affect the results as efficacy may vary with concentration. Another notable difference from clinical practice is the hourly drip frequency for the first few days in clinical settings (28).

Sunada et al. also reported low *in vitro* susceptibilities of *Acanthamoeba* to propamidine, chlorhexidine, and PHMB (29). It is important to consider that *in vitro* susceptibilities to disinfectants may not necessarily reflect clinical susceptibility and therefore may not accurately predict the effect of these agents in treating AK in a clinical setting (30).

Clinical reports have indicated that despite extended courses of combination therapies, 5%–10% of AK cases remain refractory, suggesting the need for more effective compounds with reduced clinical resistance (30, 31).

One limitation of this study concerns the artificial inactivation of the *Acanthamoeba,* which raises uncertainty regarding the comparability of this process to natural inactivation mechanisms. However, results using naked *Acanthamoeba* DNA were comparable to the results obtained with the autoclave inactivation treatment.

A strength of this study includes the feasibility of the technique used. The use of a photoreactive dye as a sample pretreatment offers a practical solution, particularly in situations where conventional culture-based techniques require too much time. This technique can be combined with any PCR assay, retaining the characteristics and performance of the original assay, as long as the fragment length is sufficient for optimal PMAxx binding. Implementing this technique in a molecular laboratory requires minimal adjustments to the current workflow. In the current study, the sample pre-processing time was efficiently limited to only 50 min, of which 45 min was the incubation period. This highlights the practicality and efficiency of the technique in a laboratory setting. While mRNA targets may provide an alternative approach to assessing viability, they come with their own challenges, including issues related to RNA stability, extraction efficiency, and potential interference from RNA remnants in dead cells.

Another strength of the study is the validation of the v-PCR assay on *Acanthamoeba* trophozoites, using two *Acanthamoeba* ATCC strains and a clinical *Acanthamoeba* strain. Additionally, the v-PCR assay was validated on *Acanthamoeba* cysts using an *Acanthamoeba* ATCC strain. The validation on both trophozoites and cysts demonstrates the applicability and effectiveness of the v-PCR method for detecting *Acanthamoeba* in different life stages. The inclusion of a clinical *Acanthamoeba* strain further enhances the relevance and practicality of the v-PCR method for real-world scenarios.

A recommendation for future studies is to further expand the application of v-PCR to clinical samples. Expanding the use of v-PCR to analyze clinical samples would be valuable in assessing *Acanthamoeba* viability in real-world scenarios and clinical settings. Furthermore, monitoring clinical samples longitudinally from individual patients would be beneficial. This approach would provide a comprehensive understanding of *Acanthamoeba* viability over time and facilitate the evaluation of treatment efficacy in AK.

In conclusion, a sensitive and rapid v-PCR assay has been validated that can discriminate between viable and non-viable *Acanthamoeba* trophozoites and cysts across multiple species, adding less than 1 h to the turnaround time. This assay holds great potential for evaluating treatment efficacy in AK. The ability to distinguish viable and non-viable *Acanthamoeba* using v-PCR allows for the monitoring of treatment response and the evaluation of the success or failure of therapeutic interventions. This information is crucial in optimizing treatment strategies and improving patient outcomes. Additionally, the v-PCR offers a valuable alternative to the current labor-intensive methods used to identify amoebicidal and cysticidal compounds against *Acanthamoeba*. By providing a more efficient and reliable means of assessing drug efficacy, v-PCR contributes to streamlining the process of identifying effective treatments.

## ACKNOWLEDGMENTS

The authors gratefully acknowledge Dr. Dieuwke van Ooik (Visser Contactlenzen, the Netherlands) for providing Menicon Progent, and Dr. Jan Coremans (Pharmacy Maastricht University Medical Center+, the Netherlands) for supplying propamidine isetionate 0.1% and chlorhexidine digluconate 0.02%. We would also like to thank Mr. Erik Beuken for his assistance with typing the clinical sample.

## AUTHOR AFFILIATIONS

[1]University Eye Clinic Maastricht, Maastricht University Medical Center+, Maastricht, the Netherlands
[2]School for Mental Health and Neuroscience (MHeNs), Maastricht University, Maastricht, the Netherlands
[3]Department of Medical Microbiology, Infectious Diseases & Infection Prevention, Maastricht University Medical Center+, Maastricht, the Netherlands
[4]Care and Public Health Research Institute (CAPHRI), Maastricht University, Maastricht, the Netherlands
[5]Department of Ophthalmology, Zuyderland Medical Center, Heerlen, the Netherlands

## AUTHOR ORCIDs

J. M. J. Veugen  http://orcid.org/0000-0002-0469-5781
P. F. G. Wolffs  http://orcid.org/0000-0002-5326-3985

## AUTHOR CONTRIBUTIONS

J. M. J. Veugen, Conceptualization, Data curation, Formal analysis, Investigation, Methodology, Project administration, Validation, Visualization, Writing – original draft, Writing – review and editing | P. H. M. Savelkoul, Resources, Supervision, Writing – review and editing | R. M. M. A. Nuijts, Supervision, Writing – review and editing | M. M. Dickman, Supervision, Writing – review and editing | P. F. G. Wolffs, Conceptualization,

Methodology, Resources, Supervision, Validation, Writing – original draft, Writing – review and editing

## ADDITIONAL FILES

The following material is available online.

### Supplemental Material

**Table S1 (Spectrum01811-24-s0001.docx).** Ct values and ΔCt values of PMAxx-treated and non-PMAxx-treated samples.

### Open Peer Review

**PEER REVIEW HISTORY (review-history.pdf).** An accounting of the reviewer comments and feedback.

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
