## [Reviewer comments · Microbiology Spectrum]

Microbiology Spectrum

Enhancing Acanthamoeba Diagnostics: Rapid Detection of Viable Acanthamoeba Trophozoites and Cysts using Viability PCR Assay

Judith Veugen, Paul Savelkoul, Rudy Nuijts, Mor Dickman, and Petra Wolffs

Corresponding Author(s): Petra Wolffs, Maastricht Universitair Medisch Centrum+

Review Timeline:

Submission Date:	September 9, 2024
Editorial Decision:	October 22, 2024
Revision Received:	January 10, 2025
Accepted:	January 13, 2025

Editor: Denis Sereno

Reviewer(s): Disclosure of reviewer identity is with reference to reviewer comments included in decision letter(s). The following individuals involved in review of your submission have agreed to reveal their identity: Pieter W Smit (Reviewer #1); Mehmet Aykur (Reviewer #2)

Transaction Report:

DOI: <https://doi.org/10.1128/spectrum.01811-24>

Re: Spectrum01811-24 (Enhancing Acanthamoeba Diagnostics: Rapid Detection of Viable Acanthamoeba Trophozoites and Cysts using Viability PCR Assay)

Dear Dr. Petra F.G. Wolffs:

Thank you for the privilege of reviewing your work. Below you will find my comments, instructions from the Spectrum editorial office, and the reviewer comments.

Revision Guidelines

Sincerely,
Denis Sereno
Editor
Microbiology Spectrum

Reviewer #1 (Comments for the Author):

J. Veugen et al describes a method developed to distinguish viable vs total DNA of acanthamoeba species via inactivation using PMAxx and detection via PCR.

The manuscript is well written in a clear manner guiding the reader in a topic in which there are not so many experts around and thus valuable to provide a bit more context.

the assay is sensitive to distinguish differences in %viable acant. (table S1) when percentage drops from 100 to 10 to 1%. This is due to the nature of PCR. Would this limitation have any clinical implications when used in routine practise? perhaps good to mention this in the discussion.

The authors decide to use PMAxx but could perhaps also have chosen a RNA target to evaluate viable acanthamoeba? It would be good if the discussion contains a section of alternative methods and why PMAxx would be a better option (or not).

minor: figures are not understandable when printed in black and white (perhaps old fashioned habit of the reviewer).

page2,line33 it does not streamline identification proces. a regular PCR is easier.

Results 205,213,221; it took me multiple reads/attempts to understand the sentence and given that you use 2 decimals written text and in table only 1 decimal, I got confused. Please make this a bit easier to understand.

Reviewer #2 (Comments for the Author):

Reviewer Comments to Author

Acanthamoeba is one of the most common free-living amoeba, especially in environmental habitats such as water, soil and dust. Acanthamoeba keratitis is more common in contact with contaminated water and soil, especially in contact lens wearers. Although Acanthamoeba has been isolated from almost everywhere in the world, it is more commonly reported in developed countries with higher case rates. This is based on the fact that there are laboratories with the necessary infrastructure to diagnose Acanthamoeba keratitis.

The aim of this study was to 'develop and validate a sensitive viability PCR (v-PCR) assay for the rapid identification of viable Acanthamoeba trophozoites and cysts, using a photoreactive dye to distinguish viable Acanthamoeba from non-viable Acanthamoeba. 'Although it is titled 'Discrimination of Viable Acanthamoeba castellani Trophozoites and Cysts by Propidium Monoazide Real-Time Polymerase Chain Reaction', there is no difference between this study and the previous study with this title.

In this study, the authors evaluated the effects of Propamidine, which is used for therapeutic purposes, and Chlorhexidine with the developed vPCR method. However, similar things were done in the other study.

There are some deficiencies in the following points.

- Why is propidium monoazide (PMAxx) not abbreviated as (PMA).
- Line 24-25: Isn't corneal scraping again to determine the efficacy of the treatment an invasive method that may limit its traceability?
- How do you evaluate this situation, since it would not be appropriate to repeat such an invasive method on the patient to evaluate the effect of the drug on the corneal scraping sample taken from the patient?
- Line 86-87: The purpose of the PCR method is to diagnose AK and to start the most effective drugs against Acanthamoeba more quickly instead of other antimycotic, antiviral and antibacterial treatments. In addition, the effectiveness of the treatment is evaluated by ophthalmologists.
- Methods.
- There are differences between the primers in the PCR method used in this study and the primers referred to. It is also important to evaluate the PCR amplification efficiency.

Results:

- The standard curve and amplification efficiency should be mentioned in the study.
- In line 206, "5.69 (1%)", a minus should be added

Reviewer Comments to Author

Acanthamoeba is one of the most common free-living amoeba, especially in environmental habitats such as water, soil and dust. *Acanthamoeba* keratitis is more common in contact with contaminated water and soil, especially in contact lens wearers. Although *Acanthamoeba* has been isolated from almost everywhere in the world, it is more commonly reported in developed countries with higher case rates. This is based on the fact that there are laboratories with the necessary infrastructure to diagnose *Acanthamoeba* keratitis.

The aim of this study was to ‘develop and validate a sensitive viability PCR (v-PCR) assay for the rapid identification of viable *Acanthamoeba* trophozoites and cysts, using a photoreactive dye to distinguish viable *Acanthamoeba* from non-viable *Acanthamoeba*. ‘Although it is titled ‘Discrimination of Viable *Acanthamoeba castellanii* Trophozoites and Cysts by Propidium Monoazide Real-Time Polymerase Chain Reaction’, there is no difference between this study and the previous study with this title.

In this study, the authors evaluated the effects of Propamidine, which is used for therapeutic purposes, and Chlorhexidine with the developed vPCR method. However, similar things were done in the other study.

There are some deficiencies in the following points.

- Why is propidium monoazide (PMAxx) not abbreviated as (PMA).
- Line 24-25: Isn't corneal scraping again to determine the efficacy of the treatment an invasive method that may limit its traceability?
- How do you evaluate this situation, since it would not be appropriate to repeat such an invasive method on the patient to evaluate the effect of the drug on the corneal scraping sample taken from the patient?
- Line 86-87: The purpose of the PCR method is to diagnose AK and to start the most effective drugs against *Acanthamoeba* more quickly instead of other antimycotic, antiviral and antibacterial treatments. In addition, the effectiveness of the treatment is evaluated by ophthalmologists.
- **Methods.**
- There are differences between the primers in the PCR method used in this study and the primers referred to. It is also important to evaluate the PCR amplification efficiency.

Results:

- The standard curve and amplification efficiency should be mentioned in the study.
- In line 206, "5.69 (1%)", a minus should be added

Response to reviewer comments

Reviewer #1 (Comments for the Author):

J. Veugen et al describes a method developed to distinguish viable vs total DNA of acanthamoeba species via inactivation using PMAxx and detection via PCR.

The manuscript is well written in a clear manner guiding the reader in a topic in which there are not so many experts around and thus valuable to provide a bit more context.

1. The assay is sensitive to distinguish differences in %viable acant. (table S1) when percentage drops from 100 to 10 to 1%. This is due to the nature of PCR. Would this limitation have any clinical implications when used in routine practise? perhaps good to mention this in the discussion.

Response: We thank the reviewer for this suggestion and added the following text to the discussion: “However, it is important to consider that the assay's sensitivity to detect small differences in viability, particularly at lower percentages of viable cells, may have implications for its clinical utility.” (Discussion, lines 268-270)

2. The authors decide to use PMAxx but could perhaps also have chosen a RNA target to evaluate viable acanthamoeba? It would be good if the discussion contains a section of alternative methods and why PMAxx would be a better option (or not).

Response: While mRNA targets could offer an alternative approach to assessing viability, they may come with their own set of challenges, including issues related to RNA stability, extraction efficiency, and potential interference from RNA remnants in dead cells.

Based on prior experiences of our research group with PMAxx treatment in evaluating viability for *Chlamydia trachomatis* (Janssen et al.) and SARS-CoV-2 (Veugen et al.), we have demonstrated that PMAxx treatment, when coupled with DNA- and RNA-based PCR methods, is easily implementable within routine diagnostics. Our previous research has demonstrated the efficacy of PMAxx in penetrating dead cells and covalently binding to both DNA and RNA, providing a reliable, well-established approach for distinguishing viable from non-viable cells in a reproducible manner."

The following text was added to the discussion: “While mRNA targets may provide an alternative approach to assessing viability, they come with their own challenges, including issues related to RNA stability, extraction efficiency, and potential interference from RNA remnants in dead cells.” (Discussion, lines 315-317)

3. minor: figures are not understandable when printed in black and white (perhaps old fashioned habit of the reviewer).

Response: We appreciate the reviewer's feedback regarding the potential challenges in interpreting the figures when printed in black and white. In response to this concern, we will ensure that the figures are published in color to enhance their clarity and comprehensibility for readers.

4. page2,line33 it does not streamline identification proces. a regular PCR is easier.

Response: The reviewer's feedback was considered, leading to the removal of this part from the sentence. The revised version now reads as follows: “By effectively distinguishing between viable and non-viable *Acanthamoeba*, this test enables monitoring of treatment response and efficacy, essential for guiding clinical interventions in AK cases.” (Importance, lines 32-34)

5. Results 205,213,221; it took me multiple reads/attempts to understand the sentence and given that you use 2 decimals written text and in table only 1 decimal, I got confused. Please make this a bit easier to understand.

We acknowledge the reviewer’s comment and have revised the presentation of data in the table to include two decimal places to match the written text. Additionally, we have revised the relevant sentences to enhance readability: “In cultured *A. polyphaga* trophozoites at a concentration of 10^5 cells/mL, mixtures showed mean delta Ct values of -1.09 (100%), -0.77 (50%), -2.75 (10%), -5.69 (1%), -8.81 (0.1%), and -12.75 (0%). These values correspond to mean log reductions of 0.33, 0.23, 0.83, 1.71, 2.64, and 3.82, respectively.” (Methods, lines 210-213, 217-220, and 225-228)

Reviewer #2 (Comments for the Author):

1. *Acanthamoeba* is one of the most common free-living amoeba, especially in environmental habitats such as water, soil and dust. *Acanthamoeba* keratitis is more common in contact with contaminated water and soil, especially in contact lens wearers. Although *Acanthamoeba* has been isolated from almost everywhere in the world, it is more commonly reported in developed countries with higher case rates. This is based on the fact that there are laboratories with the necessary infrastructure to diagnose *Acanthamoeba* keratitis.

The aim of this study was to 'develop and validate a sensitive viability PCR (v-PCR) assay for the rapid identification of viable *Acanthamoeba* trophozoites and cysts, using a photoreactive dye to distinguish viable *Acanthamoeba* from non-viable *Acanthamoeba*. 'Although it is titled 'Discrimination of Viable *Acanthamoeba castellanii* Trophozoites and Cysts by Propidium Monoazide Real-Time Polymerase Chain Reaction', there is no difference between this study and the previous study with this title.

In this study, the authors evaluated the effects of Propamidine, which is used for therapeutic purposes, and Chlorhexidine with the developed vPCR method. However, similar things were done in the other study.

Response: We appreciate the reviewer's thorough assessment and recognize the importance of comparing our study to previous research. In response to the concerns raised, we would like to highlight additional points that differentiate our work and underscore the value of conducting comprehensive studies.

1. Improved methodology: Our study introduces improvements such as the use of PMAxx, an improved version of PMA, and an expanded range of *Acanthamoeba* strains.
2. Expanded strain selection: In our study, we also included *A. polyphaga* trophozoites and cysts, which are known to be responsible for the majority of reported cases of AK worldwide along with *A. castellanii*. In addition, we tested a clinical isolate that was identified as *A. castellanii*.
3. Diversified assays:
 - We evaluated the efficacy of Menicon Progent intensive cleaner, which is a protein removing and disinfectant solution for hard lenses. Although the study of Fittipaldi et al. also evaluated the effectiveness of several contact lens disinfectants, Menicon

Progent was not included. Since Menicon Progent was the only one shown to be effective against SARS-CoV-2 in our previous research, we were interested in its activity against *Acanthamoeba* spp.

- In addition, we tested the efficacy of primary treatment options for AK; propamidine (Brolene), chlorhexidine, and a combination of these agents.

4. Emphasis on reproducibility: We recognize the importance of reproducibility and consistency in scientific research. Therefore, we believe that conducting multiple studies to validate our findings and assess their reproducibility under different conditions is crucial. This approach ensures the reliability and generalizability of our results.

These improvements contribute to the robustness and applicability of our viability PCR assay. The following text was added to the discussion: “Conducting multiple studies to validate findings and assess reproducibility under different conditions is crucial. This approach ensures the reliability and generalizability of the results.” (Discussion, lines 271-272) in addition to the paragraph that differentiates this study from the study of Fittipaldi et al. (Discussion, lines 273-302)

There are some deficiencies in the following points.

2. Why is propidium monoazide (PMAxx) not abbreviated as (PMA).

Response: PMAxx is an improved version of PMA and is therefore not abbreviated as “PMA” to avoid possible confusion arising from the distinction between the two versions.

3. Line 24-25: Isn't corneal scraping again to determine the efficacy of the treatment an invasive method that may limit its traceability?

Response: While corneal scraping is indeed an invasive method, it remains a valuable diagnostic tool in cases where direct assessment of the affected tissue is necessary for treatment monitoring and decision-making. Ensuring proper treatment is crucial in preventing disease progression and potential complications with far-reaching consequences.

4. How do you evaluate this situation, since it would not be appropriate to repeat such an invasive method on the patient to evaluate the effect of the drug on the corneal scraping sample taken from the patient?

Response: If repeated corneal scraping is considered inappropriate for the patient, another option may be to obtain an eye swab (eSwab). With an eSwab, the eye is gently swabbed to collect samples for analysis. This is less invasive than corneal scraping, but can still provide valuable information for evaluating the effect of the drug on the eye. This method can provide a less invasive alternative while still allowing the impact of the treatment on the eye condition to be assessed.

5. Line 86-87: The purpose of the PCR method is to diagnose AK and to start the most effective drugs against Acanthamoeba more quickly instead of other antimycotic, antiviral and antibacterial treatments. In addition, the effectiveness of the treatment is evaluated by ophthalmologists.

Response: We appreciate the reviewer's feedback and added the following sentence: "Ophthalmologists play a crucial role in evaluating the effectiveness of the treatment regimen to ensure optimal management of AK." (Introduction, lines 87-89)

Methods.

6. There are differences between the primers in the PCR method used in this study and the primers referred to. It is also important to evaluate the PCR amplification efficiency.

Response: We appreciate the reviewer's insightful observation regarding the differences in primer sequences and the importance of evaluating PCR amplification efficiency. In response to this feedback, we have included the following sentence in the manuscript: "Compared to the referenced publication, different concentrations were used and the reverse primer was slightly modified." (Methods, lines 191-192)

Results:

7. The standard curve and amplification efficiency should be mentioned in the study.

Response: We thank the reviewer for the feedback regarding the inclusion of the standard curve and amplification efficiency in the study. We acknowledge this suggestion and added the following sentence to the manuscript: "The amplification efficiency of the PCR reaction was 100% based on the standard curve, with a corresponding slope of -3.32 and R^2 of 0.98." (Results, lines 201-202)

8. In line 206, "5.69 (1%)", a minus should be added

Response: We appreciate the reviewer's attention to detail and suggestion for improvement regarding line 206. We have added a minus sign to accurately reflect the value as "-5.69" in the revised version of the manuscript. (Results, line 211)

Re: Spectrum01811-24R1 (Enhancing Acanthamoeba Diagnostics: Rapid Detection of Viable Acanthamoeba Trophozoites and Cysts using Viability PCR Assay)

Dear Dr. Petra F.G. Wolffs:

Your manuscript has been accepted, and I am forwarding it to the ASM production staff for publication. Your paper will first be checked to make sure all elements meet the technical requirements. ASM staff will contact you if anything needs to be revised before copyediting and production can begin. Otherwise, you will be notified when your proofs are ready to be viewed.

Sincerely,
Denis Sereno
Editor
Microbiology Spectrum